# Genomic Characterization of *Salmonella enterica* serovar Weltevreden Associated with Human Diarrhea

Jianmin Zhang,[a,b,c,d] Zhong Peng,[e,f,g] Kaifeng Chen,[a,b,c,d] Zeqiang Zhan,[a,b,c,d] Haiyan Shen,[h] Saixiang Feng,[a,b,c,d] Hongchao Gou,[h] Xiaoyun Qu,[a,b,c,d] Mark Ziemann,[i] Daniel S. Layton,[j] Xiangru Wang,[e,f] Huanchun Chen,[e,f] Bin Wu,[e,f] Xuebin Xu,[k] Ming Liao[a,b,c,d,h]

aNational and Regional Joint Engineering Laboratory for Medicament of Zoonoses Prevention and Control, College of Veterinary Medicine, South China Agricultural University, Guangzhou, China

bKey Laboratory of Zoonoses, Ministry of Agriculture, College of Veterinary Medicine, South China Agricultural University, Guangzhou, China

cKey Laboratory of Zoonoses Prevention and Control of Guangdong Province, College of Veterinary Medicine, South China Agricultural University, Guangzhou, China

dAnimal Infectious Diseases Laboratory, College of Veterinary Medicine, South China Agricultural University, Guangzhou, China

eState Key Laboratory of Agricultural Microbiology, College of Veterinary Medicine, Huazhong Agricultural University, Wuhan, China

fThe Cooperative Innovation Center for Sustainable Pig Production, Huazhong Agricultural University, Wuhan, China

gHubei Hongshan Laboratory, Wuhan, China

hInstitute of Animal Health, Guangdong Academy of Agricultural Sciences, Guangzhou, China

iSchool of Life and Environmental Sciences, Deakin University, Waurn Ponds Campus, Geelong, Victoria, Australia

jCommonwealth Scientific and Industrial Research Organization Health and Biosecurity, Australian Centre for Disease Prevention, East Geelong, Victoria, Australia

kShanghai Municipal Center for Disease Control and Prevention, Shanghai, China

Jianmin Zhang and Zhong Peng contributed equally to this work. The order was determined based on authors' contributions.

**ABSTRACT** *Salmonella* Weltevreden is an emerging pathogen associated with human diarrhea, and knowledge of the genomics and epidemiology of this serovar is still limited. In this study, we performed whole-genome sequencing of 96 *S.* Weltevreden isolates recovered from diarrheal patients and 62 isolates from food animals in China between 2006 and 2017. Together, with an additional 199 genome sequences of *S.* Weltevreden published in NCBI, we performed an analysis on all 357 *S.* Weltevreden genome sequences. Our results demonstrated that the majority of *S.* Weltevreden from diarrheal patients from China (97.92%, 94/96) and the other regions in the world (94.97%, 189/199) identified in this study were sequence type (ST) 365. The remaining types were ST3771 ($n = 3$), ST22 ($n = 1$), ST155 ($n = 1$), and ST684 ($n = 1$). In addition, ST365 was also widely recovered from animals, food, and environmental samples in different regions of the world. Phylogenetic analysis and pulsed-field gel electrophoresis (PFGE) revealed that *S.* Weltevreden from diarrheal patients was closely related to those recovered from food and environmental specimens. We also showed that *S.* Weltevreden did not exhibit severe antimicrobial resistance profiles, suggesting administering antibiotics is still effective for controlling the agent. Interestingly, we found that *S.* Weltevreden strains carried a number of virulence factor genes, and a 100.03-kb IncFII(S) type plasmid was widely distributed in *S.* Weltevreden strains. Elimination of this plasmid decreased the bacterial capacity to infect both Caco-2 cells and C57BL/6 mice, suggesting the importance of this plasmid for bacterial virulence. Our results contribute to the understanding of the epidemiology and virulence of *S.* Weltevreden.

**IMPORTANCE** *Salmonella* Weltevreden is a pathogen associated with human diarrheal diseases found across the globe. However, knowledge of the genomics and epidemiology of this pathogen is still limited. In this study, we found *S.* Weltevreden sequence type (ST) 365 is commonly recovered from diarrheal patients in China and many other regions of the world, and there is no major difference between the Chinese isolates and the global isolates at the phylogenetic level. We also demonstrated that ST365 was widely recovered from animal, food, and environmental samples collected in different,

Address correspondence to Bin Wu, wub@mail.hzau.edu.cn, Xuebin Xu, xuxuebin@scdc.sh.cn, or Ming Liao, mliao@scau.edu.cn.

The authors declare no conflict of interest.

global regions. Importantly, we discovered an IncFII(S) type plasmid commonly carried by *S.* Weltevreden strains of human, animal, and food origins, and this plasmid is likely to contribute to the bacterial pathogenesis. These findings enhance our understanding of the emergence of *S.* Weltevreden involved in diarrheal outbreaks and the global spread of *S.* Weltevreden strains.

**KEYWORDS** IncFII(S) type plasmid, ST365, *Salmonella* Weltevreden, human diarrhea, whole-genome sequence and phylogenetic analysis

*S*almonella has been identified as 1 of the 4 key causes of human diarrheal diseases, and Salmonellosis is the third leading cause of death among diarrheal diseases worldwide (1, 2). According to the estimates of the Centers for Disease Control and Prevention, *Salmonella* causes approximately 1.35 million infections, 26,500 hospitalizations, and 420 deaths in the United States every year (3). In China, *Salmonella* is one of the most harmful foodborne pathogens; several published articles document that 70 to 80% of bacterial food-poisoning incidents are caused by *Salmonella* (4–6). A recent study analyzed bacterial foodborne diseases in China between 2003 and 2017 found *Salmonella* was associated with 899 outbreaks, which resulted in 11,351 hospitalizations and 4 deaths (7). Consumption of contaminated food, particularly food of animal origin such as eggs, meat, poultry, and milk, are the main sources of *Salmonella* infections. This is due to the high prevalence of *Salmonella* in animals, particularly in food animals such as poultry, pigs, and cattle (1, 8).

To date, *Salmonella* strains are classified into 6 subspecies containing over 2,500 serovars (9). Several serovars, such as Typhimurium, Enteritidis, and Derby, are well-known as they are frequently detected in China and the other parts of the world (10–14). For example, a recent study analyzed 35,382 serotyped *Salmonella* in China between 1982 and 2019, and found Typhimurium was the most dominant serovar in the country, followed by Enteritidis and Derby (10). In addition, the proportion of Typhimurium and other serovars, including London, Rissen, Corvallis, Meleagridis, Kentucky, and Goldcoast, displayed an increasing prevalence from 2006 to 2019 (10). Apart from these common serovars, several new serovars, such as Telkebier (15), Uzaramo (16), Changwanni (17), and Weltevreden (18), have been also detected in China in recent years, which complicates surveillance and increases the risk of disease-outbreaks caused by these new serovars. Of particular note is *Salmonella* Weltevreden, which has caused several outbreaks around the world, including Réunion Island (19), Europe (e.g., Norway, Denmark, and Finland) (20), and Southeast Asia (e.g., India, Malaysia, Thailand, and Laos) (18, 21–26). In India, a foodborne outbreak of *S.* Weltevreden ST1500 caused an acute, watery, diarrheal illness in 150 students aged between 20 to 30 years (27). In China, 40 diarrhea cases due to *S.* Weltevreden infections in southern coastal areas have been reported, and *S.* Weltevreden isolates recovered from these cases were determined as ST365 (18). More recently, an investigation of a large, national, multiple-serotype *Salmonella* outbreak linked to contaminated kratom in the United States between 2017 and 2018 characterized 6 *S.* Weltevreden strains (28). These findings indicate that *S.* Weltevreden may represent a threat to human health, though comprehensive knowledge of genotypes, epidemiology, and biogeography of *S.* Weltevreden associated with human diarrhea is still unclear. In addition, *S.* Weltevreden has been detected or isolated from animal, food, and environmental samples (29–31). Therefore, continuously monitoring the distribution and genomic characteristics of *S.* Weltevreden strains from both humans and samples from the other hosts, and analyzing the association between human and non-human isolates is of great significance. In this study, we performed whole-genome sequencing (WGS) of 96 *S.* Weltevreden strains isolated from diarrheal patients and 62 strains isolated from different types of animals or foods. Together, with an additional 199 published genome sequences, we generated a comprehensive analysis of 357 *S.* Weltevreden genome sequences. Our aim is to understand the molecular basis for the virulence and pathogenesis of the emerging *S.* Weltevreden.

## RESULTS

**ST365 is the predominant sequence type of *Salmonella* Weltevreden strains isolated from diarrheal patients, and food-related samples.** Between 2006 and 2017, we isolated and stored *S.* Weltevreden strains (*n* = 118) from the diarrheal stool and blood of 118 diarrheal patients admitted to hospitals in Shanghai (20 patients), Guangdong (59 patients), Guangxi (38 patients), and Yunnan (1 patient) in China (Data set S1). However, only 96 of these strains were able to be recovered from the laboratory stocks successfully. Therefore, these 96 strains were used for Illumina sequencing and other analyses in this study (Data set S2). Our initial pulsed-field gel electropohoresis (PFGE) typing results revealed that many isolates recovered in Guangdong and Guangxi had similar PFGE types (Fig. 1). Multilocus sequence typing (MLST) assigned these 96 sequenced isolates into 3 catalogues of STs (ST365, ST684, ST155), with 94 of them determined to be ST365 (Fig. 2A). At the time of writing this paper (32), there were 199 assembled genome sequences of *S.* Weltevreden published in NCBI Genome database, including 141 from diarrheal patients (Data set S3) and 58 from non-human-associated samples (Data set S4). We then typed these 141 human *S.* Weltevreden strains through the MLST method. Strikingly, the majority of these 141 isolates (136/141) also belonged to ST365 (Fig. 2B).

We also sequenced 62 strains isolated from animals or foods (Data set S5). MLST analysis on these 62 isolates identified 3 STs (ST365, ST40, and ST241), with 60 isolates belonging to ST365 (Fig. 2A). MLST analysis of the 58 *S.* Weltevreden non-human isolates (Data set S4) in NCBI revealed that the majority of these 58 isolates (53/58) were assigned as ST365 (Fig. 2B). Geographic analysis demonstrated that *S.* Weltevreden strains, including the ST365 clone, were distributed in many regions worldwide, including Asia, Africa, Europe, North America, and Oceania (Fig. 2C). The above findings indicated that ST365 is the predominant sequence type of *S.* Weltevreden strains analyzed in this study.

***S.* Weltevreden strains isolated from diarrheal patients were closely related to those isolated from food-related samples.** To explore the relationship between *S.* Weltevreden strains isolated from different hosts and/or from different geographic regions, a maximum-likelihood phylogenetic tree was generated based on the whole-genome sequences of the 357 *S.* Weltevreden strains (including 96 human isolates and 62 non-human isolates determined in this study, as well as 141 human isolates and 58 non-human isolates from NCBI) (Fig. 3). The results revealed that *S.* Weltevreden strains isolated from diarrheal patients were closely related to those isolated from animals, seafood, and environmental samples (Fig. 3). Not surprisingly, PFGE typing showed that most of the *S.* Weltevreden isolates from patients shared the same PFGE types with those from chicken and/or chicken feces collected from the Guangdong Province, and several human isolates shared the same types with those strains recovered from pork or cucumber seeds (Fig. 1). In addition, *S.* Weltevreden strains isolated from China also displayed a close relationship with those isolated from outside China (Fig. 3). Phylogenetic analysis identified several sequence types closely related to ST365, including ST2183, ST2383, ST3771, and ST3902 (Fig. 3). The above findings indicated that food animals and their related products might be the main contamination source of *S.* Weltevreden and the ST365 clone causing human diarrhea.

***Salmonella* Weltevreden strains did not contain a particular abundance of antimicrobial resistance genes but possessed a mass of virulence factor genes.** To explore the molecular basis for the fitness and pathogenesis of *S.* Weltevreden, antimicrobial resistance genes (ARGs), and virulence factor genes (VFGs) carried by the bacterial strains were identified in this study. Identification of ARGs showed that *S.* Weltevreden strains did not contain a particular abundance of ARGs (Fig. 4A and Fig. S1). Overall, higher numbers of ARGs were identified in non-human isolates of *S.* Weltevreden than in the human isolates. ARGs commonly identified in *S.* Weltevreden strains analyzed in this study included aminoglycoside resistance genes [*aac(6′)-Iaa*, *aac(6′)-Ib-cr*, *aph(3′′)-Ib*, *aph(6)-Id*], fluoroquinolone resistance genes [*aac(6′)-Ib-cr*], sulfonamide resistance genes (*sul1*, *sul2*), and tetracycline resistance genes [*tet(A)*] (Fig. S1). Antimicrobial susceptibility testing (AST) indicated a certain association between the AMR phenotypes and the carriage of ARGs. For example, AST revealed that *S.* Weltevreden strains were susceptible to

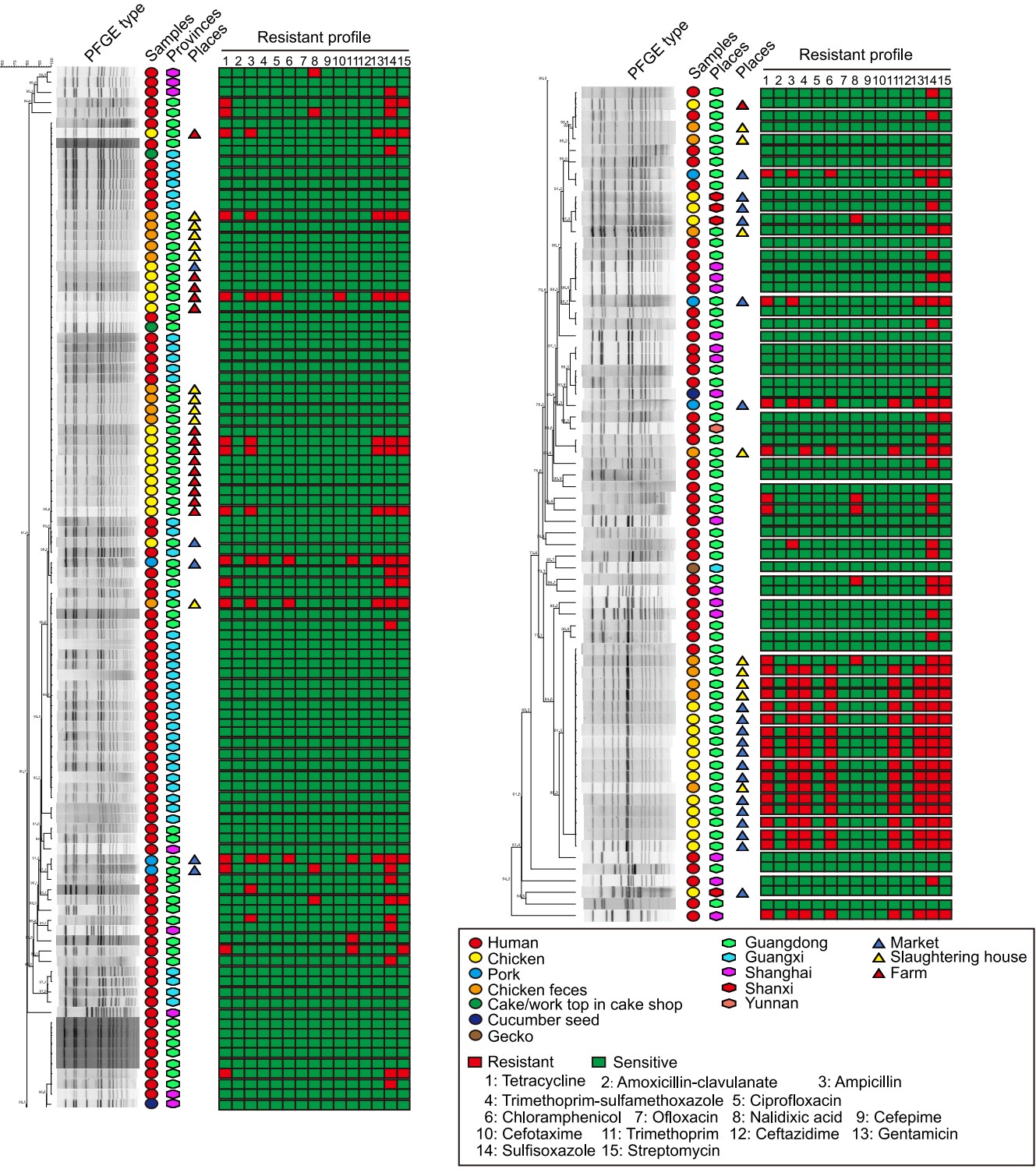

**FIG 1** PFGE patterns of *S*. Weltevreden strains from different hosts. The heat map shows the resistance profiles of individual isolates. Red indicates resistance and green indicates sensitivity. Geographic regions of the isolates are marked with hexagons; hosts are marked with circles; triangles indicate the places of non-human originated isolates.

most of the antimicrobials tested (Fig. 1). However, *S*. Weltevreden strains from animals or environmental samples displayed more severe AMR phenotypes than those strains isolated from humans (Fig. 1). Regarding resistance phenotypes to different antimicrobial classes, resistance to aminoglycosides (gentamicin, streptomycin), sulfonamides

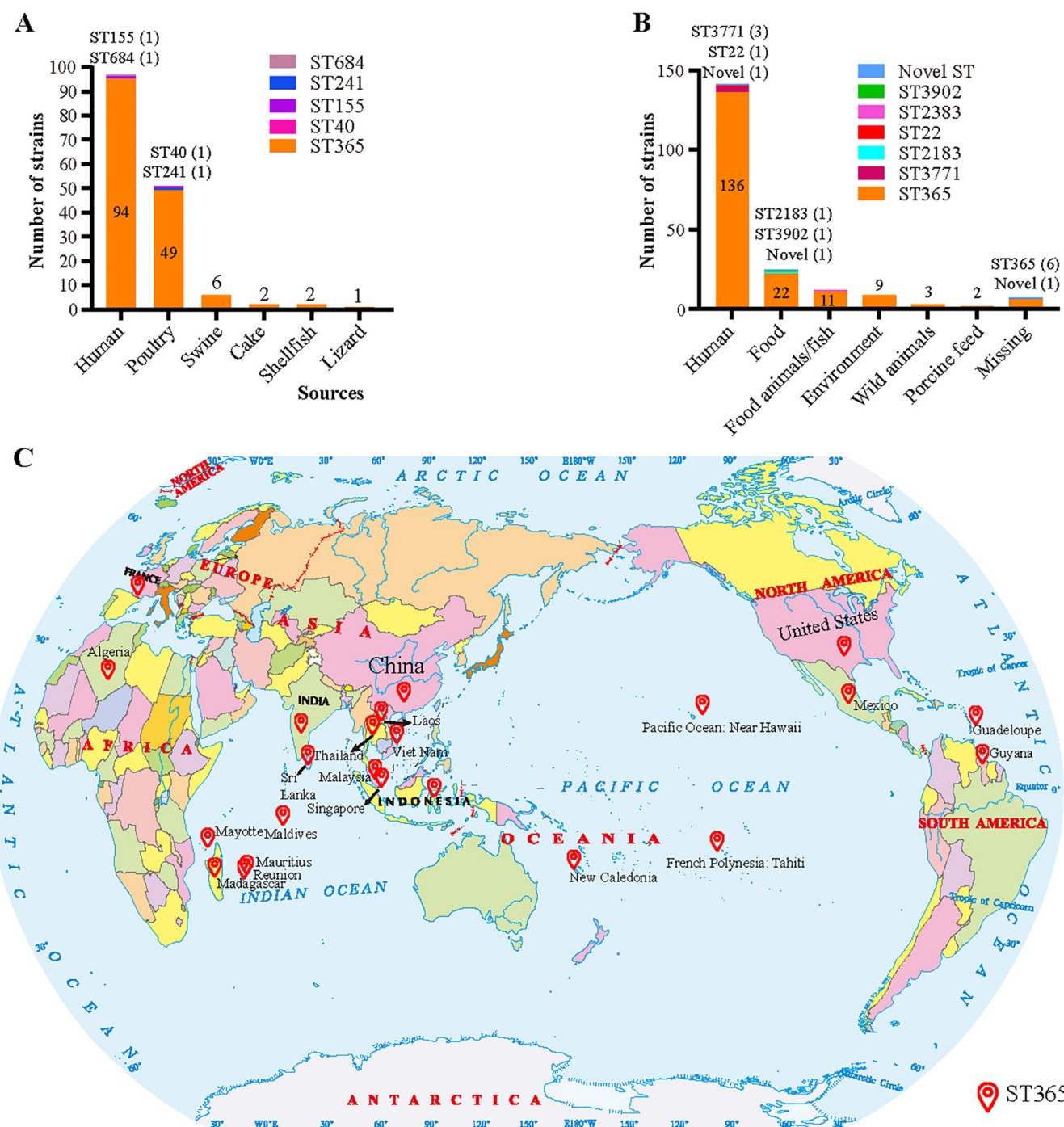

**FIG 2** Characterization of *S.* Weltevreden isolates. (A) The sequence types of the 96 sequenced *S.* Weltevreden strains recovered from diarrheal patients, and the 62 sequenced *S.* Weltevreden strains recovered from non-human species in this study. (B) The sequence types of the 199 *S.* Weltevreden strains (141 human strains and 58 non-human strains) downloaded from NCBI. (C) Shows the global distribution of *S.* Weltevreden ST365. The standard world map (no. GS[2016]2968) is provided and authorized by the Ministry of Natural Resources of the People's Republic of China for noncommercial use.

(sulfisoxazole), fluoroquinolones (nalidixic acid), and tetracyclines (tetracycline) was relatively common (Fig. 1).

Identification of VFGs showed that *S.* Weltevreden possessed a mass of VFGs, and these VFGs were involved in bacterial adherence, magnesium uptake, resistance to antimicrobial peptides, serum resistance, anti-stress, toxin production, and other bioactivities associated with the pathogenesis and fitness (Fig. 4A and Fig. S2). *In vitro* tests

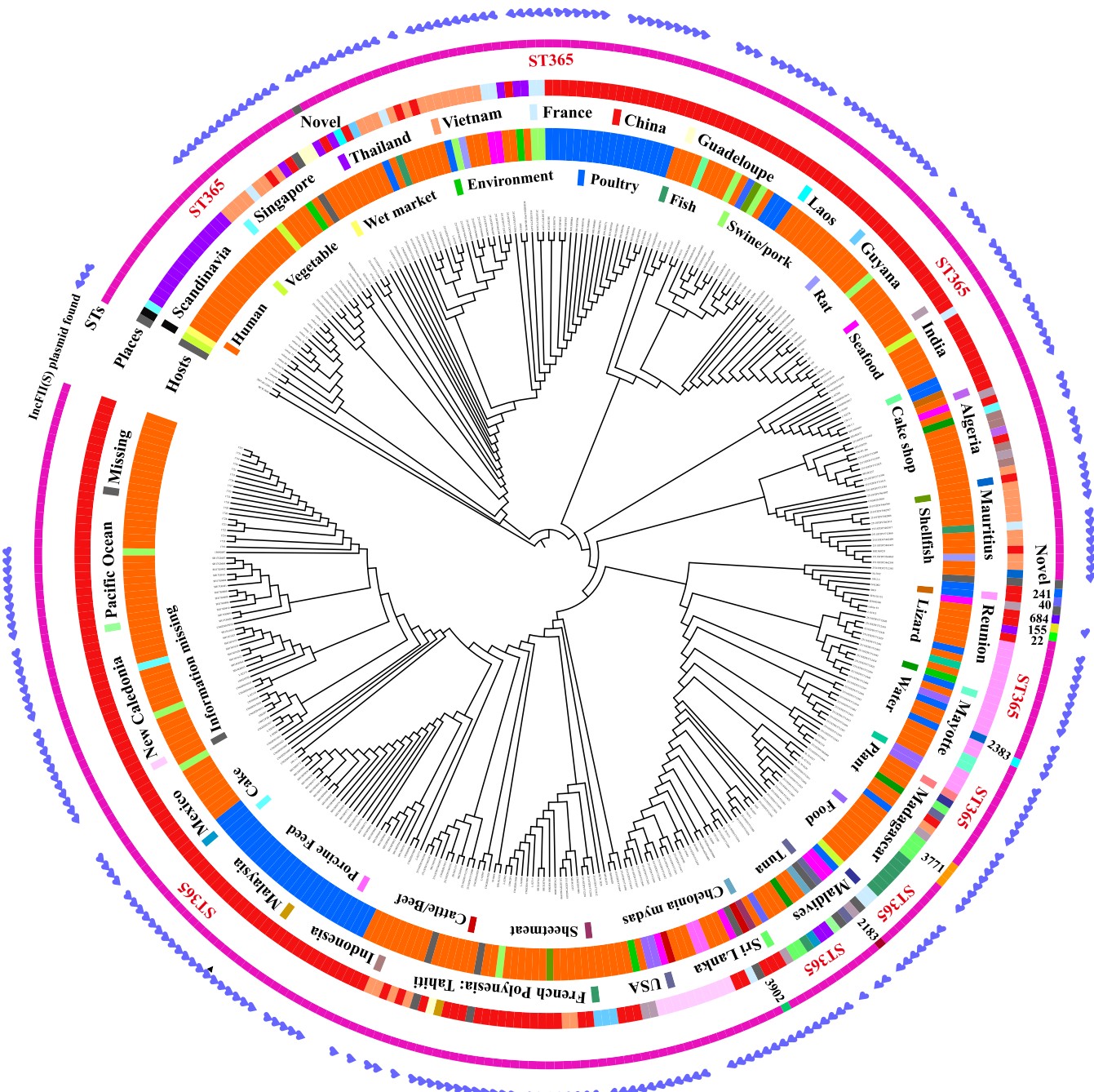

**FIG 3** Phylogenetic relationship of *S.* Weltevreden isolates (*n* = 357) from different regions, hosts, and sequence types. The maximum-likelihood tree was generated based on the single nucleotide polymorphisms across the whole-genome sequence (gSNPs) using Gubbins v2.4.0. Circles from inside to outside indicate the hosts of the isolates (circle 1), regions of isolation (circle 2), sequence types of the isolates (circle 3), and whether they carry the IncFII(S) plasmid (circle 4), respectively.

in human colon epithelial cells (Caco-2; ATCC HTB-37) showed that *S.* Weltevreden strains exhibited a strong capacity for invading Caco-2 cells compared to the virulent *S. Typhimurium* ST19 strain SL1344 (Fig. 4B), whereas *in vivo* tests in mouse models showed that administration of *S.* Weltevreden by gavage led to the loss of body weight and bacterial invasion of multiple organs in challenged mice (Fig. 4C and D).

**An IncFII(S) type plasmid was likely to be associated with the virulence of *Salmonella* Weltevreden strains.** Analysis of putative plasmids identified 22 types of different plasmid replicons (Data set S6). Strikingly, an IncFII(S) type plasmid was found in 95.04% (268/282) of the plasmid-carrying *S.* Weltevreden strains and in 95.96% (261/

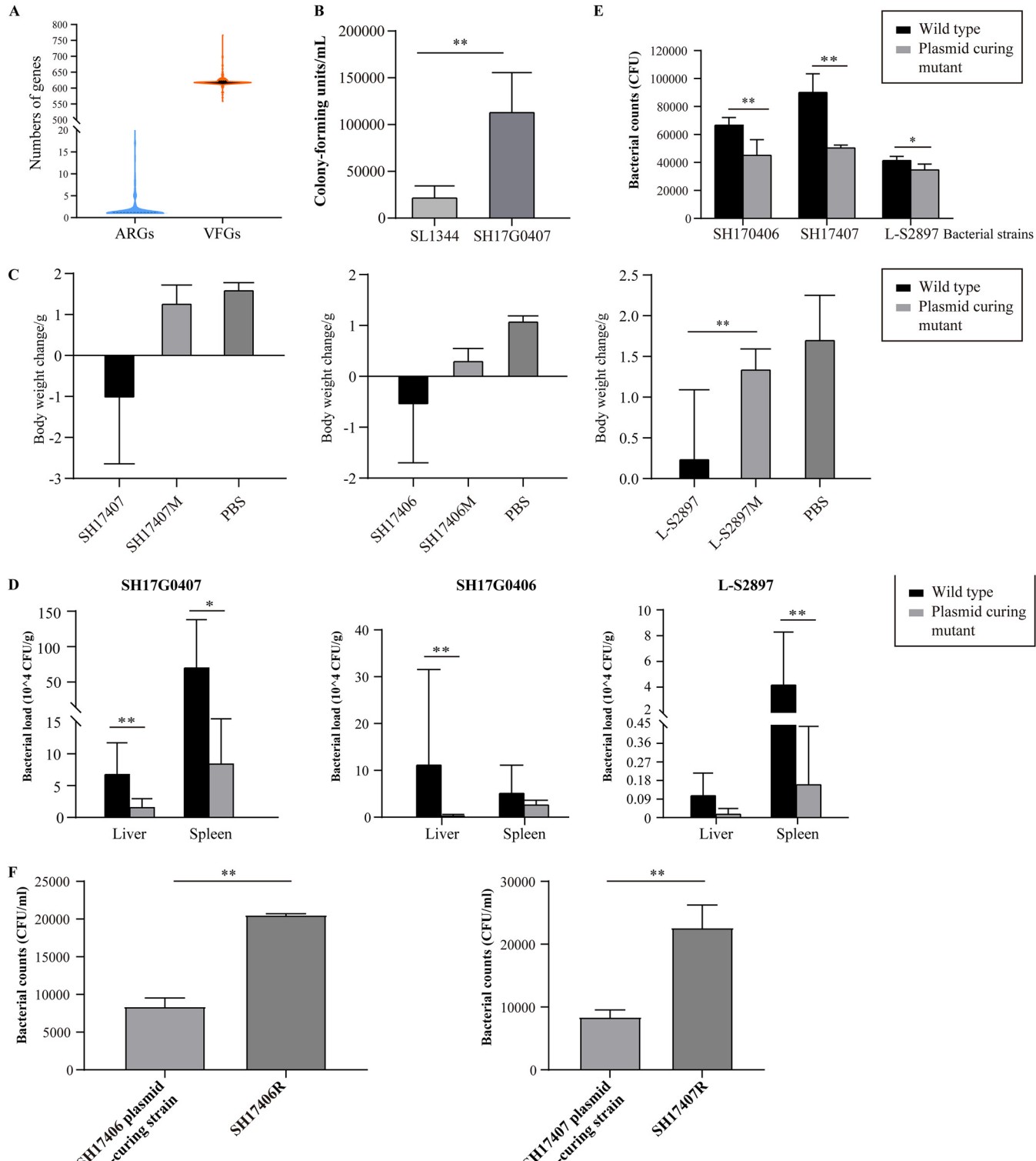

**FIG 4** Genetic characteristics and virulence of *S*. Weltevreden. (A) A violin plot displaying the numbers of ARGs and VFGs carried by *S*. Weltevreden isolates. (B) A column chart showing the capacity of *S*. Weltevreden ST365 isolate SH17G0407 from human and the *S*. *Typhimurium* virulent ST19 strain SL1344 invading to the Caco-2 cells. (C) Body weight decrease of experimental mice challenged with *S*. Weltevreden, the plasmid-curing mutant and PBS. (D) Bacterial load in the livers and spleens of experimental mice challenged with *S*. Weltevreden and the plasmid-curing mutant. (E) A column chart showing the number of *S*. Weltevreden strains and their IncFII(S)-plasmid elimination strains invading to Caco-2 cells. (F) A column chart showing the number of *S*. Weltevreden IncFII(S)-plasmid elimination strains and plasmid-complementary strains invading to Caco-2 cells. Data represents mean ± SD. The significance level was set at $P < 0.05$ (*) or $P < 0.01$ (**).

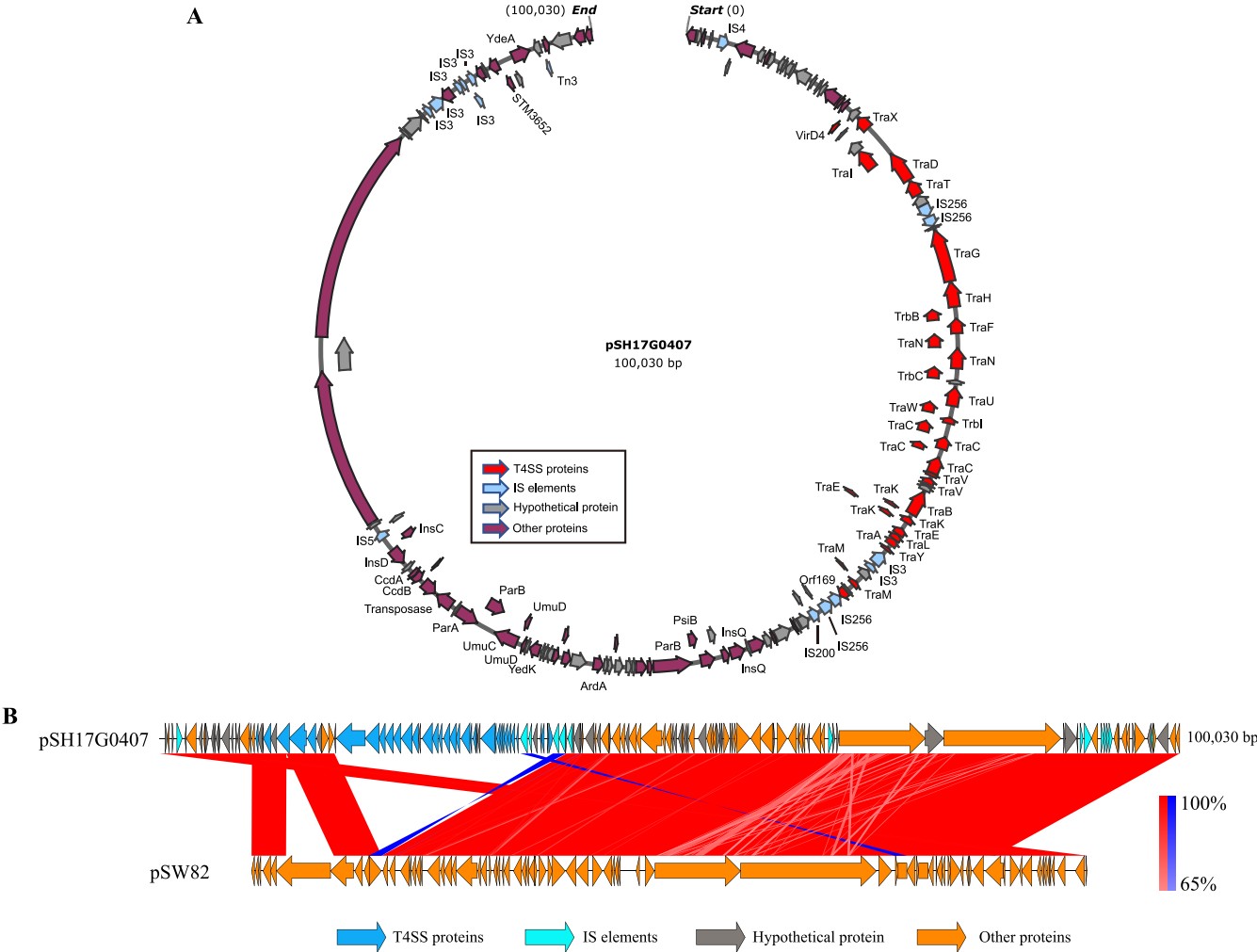

**FIG 5** Genomic characterization of the T4SS-bearing IncFII(S) type plasmid pSH17G0407. (A) A circle map of plasmid pSH17G0407 from isolate SH17G0407. Predicted coding sequences were shown used arrows in different colors (gray, genes encoding hypothetical proteins; red, genes encoding T4SS proteins). (B) Whole-genome sequence comparison of plasmids pSH17G0407 and pSW82 (GenBank accession no. NT_187135.1).

272) of the plasmid-carrying *S.* Weltevreden ST365 strains (Fig. 3). To further analyze this plasmid, we generated the complete genome sequence of the IncFII(S) type plasmid (designated pSH17G0407; GenBank accession no. MW405382) harbored in isolate SH17G0407 through Nanopore sequencing. The result showed that the complete genome sequence of pSH17G0407 was 100.03-kb in size with a G + C content of approximately 49.2% (Fig. 5A). While the complete genome sequence of this IncFII(S) type plasmid in the other *S.* Weltevreden isolates were not determined, BLASTn analysis indicated that homologous sequences of pSH17G0407 were present in the genomes of the *S.* Weltevreden strains (Data set S7). Bioinformatic analysis revealed that pSH17G0407 contained a putative type IV secretion system (T4SS) encoding region. This region included 63 genes and was flanked by 2 insertion sequences belonging to the IS*256* family (ISSod4) and the IS*4* family (ISSfl1) (Fig. 5A). Whole-genome sequence comparison revealed that pSH17G0407 was highly homologous (average nucleotide identity [ANI]: 99.93%) to a plasmid determined in *S.* Weltevreden 2007-60-3289-1 (plasmid pSW82, GenBank accession no. NT_187135.1) (Fig. 5B). To explore whether the IncFII(S) type plasmid has an impact on the bacterial virulence, the plasmids were eliminated from several randomly selected *S.* Weltevreden strains. Cell invasion assays showed that the wild type parent strains displayed a stronger invasion capacity in Caco-2 cells than the plasmid-curing mutants (Fig. 4E). We next transformed the plasmid pSH17G0407 back into the plasmid-curing mutants and assessed the bacterial invasion capacity. The results demonstrated that transforming the plasmid back

increased bacterial invasion of Caco-2 cells compared to the plasmid-curing mutants (Fig. 4F). In mouse models, challenge with wide-type bacteria caused greater, severe body weight decrease, as well as more bacterial load in different organs (liver and spleen) than the plasmid-curing mutants (Fig. 4C and D). Taken together, the above findings suggest that the IncFII(S) type plasmid might be associated with the virulence of *Salmonella* Weltevreden strains.

## DISCUSSION

In this study, we characterized the emergent *S*. Weltevreden strains associated with human diarrhea by sequencing the whole genomes of 96 *S*. Weltevreden isolates recovered from diarrheal patients in 4 provinces in China between 2006 and 2017. While *S*. Weltevreden were detected in the poultry supply chain and other meat samples in China previously (33, 34), *S*. Weltevreden strains recovered from human diarrheal cases have not been characterized in the country until recently (18). In that study, the authors determined the whole-genome sequences of 40 *S*. Weltevreden isolates from human stool samples collected between 2015 and 2016 in the Guangdong province, and found that all of these 40 isolates belonged to ST365 (18). Surprisingly, the sequence type for 94 of the 96 human isolates recovered in China in this study was also ST365. Notably, the sequence type of 136 of the 141 isolates associated with human diarrhea found globally was also ST365. These findings suggest that ST365 might represent an epidemic clone associated with human diarrhea due to *S*. Weltevreden infections. Considering there is still lack of comprehensive data for the distribution and epidemiology of *S*. Weltevreden, particularly the ST365 clone, the data presented herein provide an alarming condition of *S*. Weltevreden and its ST365 clone in the world. However, more studies are necessary to continuously monitor this *Salmonella* serovar as well as the ST365 clone.

As an important foodborne pathogen, consumption of contaminated foods is proposed to be the main cause of *Salmonella* infection in humans (2). Not surprisingly, the PFGE typing results showed that *S*. Weltevreden strains recovered from either stool samples or blood of the patients during the outbreaks in China between 2006 and 2017 had similar PFGE patterns with those isolated from poultry, pork, and/or other food types in south China. In addition, *S*. Weltevreden strains with similar PFGE types were also isolated from chicken/pig farms, slaughtering houses, and markets. These findings suggest that animals, particularly food animals and their products, might be an important source of the spread of *S*. Weltevreden to humans.

Our genomic epidemiological analyses demonstrated that *S*. Weltevreden ST365 was also widely recovered from diarrheal patients in many other regions in the world, indicating that *S*. Weltevreden ST365 is a worldwide pathogen and represents a significant risk to human health. Notably, this clone had multiple origins, including food animals, fishes, meat products, and environmental samples, suggesting food contamination should not be ignored. Before our study, a previously study analyzed the phylogeny of 115 *S*. Weltevreden isolates, and found that the population of *S*. Weltevreden could be segregated into 2 main phylogenetic clusters: one associated predominantly with continental Southeast Asia and the other more internationally dispersed (35). In this study, using a similar methodology, we performed a phylogenetic analysis of more than 350 isolates. While our results also demonstrated a similar finding that the population of 357 *S*. Weltevreden could be segregated into 2 main phylogenetic clusters, we found those 2 clusters were all internationally dispersed. With the availability of more genome sequences in the future, the population genome structure of *S*. Weltevreden will be clearer.

Knowledge of the genetic mechanisms of AMR is critical for defining appropriate treatments, refining diagnostics, and conducting epidemiological studies of AMR (36). Interestingly, our prediction of ARGs and AMR phenotype determination revealed that *S*. Weltevreden strains, including the ST365 clone, did not show a serious resistance profile, suggesting that many antimicrobial agents may still be effective for the treatment of infections caused by *S*. Weltevreden. However, multidrug-resistance phenotypes were also determined, particularly among those isolates recovered from slaughterhouses

and markets. These isolates possess a strong possibility for transmission to humans. Identification of VFGs revealed that each of the *S*. Weltevreden isolates, including the ST365 clone, possessed numerous VFGs. These VFGs encoded proteins participate in bacterial adherence, magnesium uptake, resistance to antimicrobial peptides, serum resistance, anti-stress, toxin, etc. All these bioactivities are beneficial for bacterial survival and fitness in hosts and therefore contribute to the pathogenesis (37).

Mobile genetic elements, particularly plasmids, play important roles in the dissemination of ARGs or VFGs in many bacterial species, particularly in members belonging to *Enterobacteriaceae* (38, 39). Here, we analyzed putative plasmids harbored in *S*. Weltevreden, including the ST365 clone, and found the widespread presence of an IncFII (S) type plasmid. We identified and solved the sequence of an IncFII(S) type plasmid associated with the ST365 clone and other *S*. Weltevreden strains. A notable finding is that the plasmid (pSH17G0407) determined in this study contained a putative VirB/D4 T4SS. It has been recognized that T4SSs represents major bacterial virulence determinants (40), and notably, a previous study has found that *S*. Heidelberg strains containing the VirB/D4 T4SS plasmids invaded and survived in epithelial cells and macrophages to a greater degree than those without the plasmid (41). In agreement with the findings in this study, we also found that *S*. Weltevreden strains containing the T4SS-bearing-IncFII (S) type plasmid were more invasive than the plasmid-curing strains in both Caco-2 cells and mouse models. In addition, transforming the plasmid back increased bacterial invasions to Caco-2 cells compared to the plasmid-curing mutants. These findings suggest that the T4SS-bearing-IncFII(S) type plasmid might contribute to the virulence of *S*. Weltevreden. However, further experiments are necessary to confirm this suggestion. Notably, we attempted conjugation experiments aiming to demonstrate the transfer of this IncFII(S) type plasmid pSH17G0407 into recipient *Escherichia coli* or *S. enterica* isolates, but these were unsuccessful.

In conclusion, we performed a comprehensive genome analysis of *S*. Weltevreden strains recovered from both humans and non-human species in this study. Our results revealed that *S*. Weltevreden ST365 is a worldwide clone and might represent high risks to human health. We also demonstrate that *S*. Weltevreden and the ST365 clone were abundant in food animals, food, as well as several environmental samples, and these human isolates displayed a close relationship to those recovered from food and environmental specimens. This suggests that *S*. Weltevreden, including the ST365 clone, in humans is associated with food contamination so improving food safety is necessary. Our WGS analysis showed that *S*. Weltevreden strains, including the ST365 clone, did not contain an abundance of ARGs, and *S*. Weltevreden strains were still susceptible to many antibiotics currently used, suggesting oral antibiotics remains an effective way for the treatment of *S*. Weltevreden infections. Our WGS analysis also revealed that *S*. Weltevreden carried a mass of VFGs, and a T4SS-carrying-InFII(S) plasmid was widely presented in *S*. Weltevreden strains and the ST365 clone. This InFII(S) plasmid is likely to be associated with virulence of *S*. Weltevreden and the ST365 clone. Our framework presented herein will facilitate future studies investigating the emergence of *S*. Weltevreden involved in human diarrhea and the global spread of *S*. Weltevreden strains.

## MATERIALS AND METHODS

**Bacterial strains and whole-genome sequences.** *S*. Weltevreden strains analyzed in this study included 96 strains isolated from the stool, blood, and the residual food samples of diarrheal patients admitted in hospitals in China between 2006 and 2017 (Data set S2), and 62 strains isolated from different types of animals or foods (Data set S5). In addition, the whole-genome sequences of 199 *S*. Weltevreden strains from different regions in the world were downloaded from the NCBI Genome database (Data sets S3 and 4) (32). The hosts, places, and times of isolation of these 199 *S*. Weltevreden strains were collected from their biosample registrations published in NCBI.

**Antimicrobial susceptibility testing.** Antimicrobial susceptibility testing was conducted through the broth microdilution methods, as recommended by the United States Clinical and Laboratory Standards Institute (CLSI M31-S1). A total of 15 types of antimicrobials belonging to aminoglycosides (gentamicin, streptomycin), beta-lactams (amoxicillin-clavulanate, ampicillin, cefepime, cefotaxime, ceftazidime), phenicols (chloramphenicol), trimethoprim, fluoroquinolones (ciprofloxacin, ofloxacin, nalidixic acid), sulfonamides

(trimethoprim-sulfamethoxazole, sulfisoxazole), and tetracyclines (tetracycline) were tested. To determine the MIC values, the initial concentration of each type of the antibiotics was prepared as 512 $\mu$g/mL. Results were interpreted using the CLSI breakpoints (CLSI M100, 28th Edition). Each of the antibiotics was tested with 3 duplicates. For quality control, *E. coli* ATCC 25922 was used.

**PFGE.** PFGE was performed by following the standardized protocol used by PulsedNet participating laboratories (42). Briefly, genomic DNA of each of the isolates were digested using the restriction enzyme XbaI and was then analyzed using PFGE, as described previously (43). *Salmonella* H9812 was used as a standard control strain. A molecular Imager Gel Doc XR System Universal Hood II (Bio-Rad Laboratories) was used to generate the PFGE gel pictures. Results were analyzed using the Bionumerics software (Version 5.1; Applied-Maths).

**WGS and data availability statement.** The draft genome sequences of *S.* Weltevreden isolates were generated using Illumina sequencing. To prepare high-quality sequencing libraries, we used a commercial TIANamp Bacteria DNA Kit (TIANGEN) to extract genomic DNA, and analyzed the DNA quality and quantity by electrophoresis on a 1% agarose gel, as well as a Qubit 2.0 (Thermo Scientific). DNA libraries were generated using a NEBNext Ultra II DNA Library Prep Kit (New England BioLabs), and were then sequenced on an Illumina NovaSeq 6000 platform (Illumina Inc.) at Novogene Co. LTD, using the pair-end 350 bp sequencing protocol. A total of 3.32 to 5.62 Mb raw reads were yielded for the 158 *S.* Weltevreden strains. Potential interspecies and cross-species contamination in the whole genome sequence data was checked with ConFindr (44) and QC-Blind (45). Raw reads with low quality base pairs at each terminal (Quality-Value < 20), and/or those with a short length (parameter setting at 50 bp), or > 15 bp overlap with Illumina TruSeq adapter sequences (parameter setting at 15 bp) were removed. After filtering, approximately 3.32 to 5.61 clean reads (Q20% $\geq$ 97.22%) were produced. These high-quality reads were *de novo* assembled using SPAdes (version 3.9.0) (46) to generate contigs.

We next generated the complete sequence of a plasmid (pSH17G0407) present in *S.* Weltevreden isolate SH17G0407 using Oxford Nanopore technology (ONT) in combination with the Illumina technology. Plasmid DNA was extracted using the phenol-chloroform protocol combined with Phase Lock Gel tubes (Qiagen GmbH), and was detected by the agarose gel electrophoresis as well as quantified by Qubit 2.0 (Thermo Scientific). Libraries for ONT sequencing were prepared using an SQK-LSK109 kit of Oxford Nanopore Technologies Company, while libraries for Illumina sequencing were prepared by using a NEBNext Ultra DNA Library Prep Kit for Illumina (NEB) following the manufacturer's instructions. Prepared DNA libraries were sequenced using Nanopore PromethION platform and Illumina NovaSeq PE150 at Novogene Co. LTD, respectively. ONT and Illumina short reads were finally assembled and combined using the Unicycler v0.4.7 software with default parameters.

**Bioinformatic analyses.** To determine bacterial STs, the whole-genome sequences were submitted against the PubMLST *Salmonella* typing database (https://pubmlst.org/bigsdb?db=pubmlst_salmonella_seqdef). Genome sequences were annotated by using the RAST server (47). ARGs and VFGs were determined by ResFinder 4.0 (threshold for 90% identity plus minimum 60% coverage) (https://cge.food.dtu.dk/services/ResFinder/) and VFanalyzer (threshold for 90% identity plus minimum 60% coverage) in the VFDB database (threshold for minimum 95% identity plus minimum 60% coverage) (48), respectively. Plasmids were typed using PlasmidFinder 2.1 (49). Presence of T4SS proteins and insertion elements were determined using SecReT4 2.0 (50) and IS finder (51), respectively. Phylogenetic analysis was performed as described previously (35). Briefly, the whole-genome sequences of each of the 357 *S.* Weltevreden strains were aligned against the reference genome (strain 3) using MUMmer v3.1 to obtain all potential single nucleotide polymorphism (SNP) loci (52). Indels and adjacent mismatches were removed as they are not considered as true SNPs (53). Recombination in the filtered multi-FASTA alignment was checked using Gubbins (version 1.3.4) (54). A maximum-likelihood phylogenetic tree based on the obtained SNPs was constructed using RAxML (version 8) (55) with the GAMMA GTR model. The obtained phylogenetic tree was visualized with the iTOL online tool (56).

**Plasmid elimination assay and plasmid transformation assay.** An IncFII(S) type plasmid determined in most of the *S.* Weltevreden strains in this study was eliminated by using the ethidium bromide (EB) protocol as described previously (57). Briefly, a small inoculum (approximately $10^4$ CFU/mL) of *S.* Weltevreden were grown in LB broth (Sigma-Aldrich) containing 30 $\mu$g/mL ethidium bromide (EB) until slight turbidity was observed. Afterwards, bacterial culture with appropriate dilution was plated on LB agar and incubated at 37°C overnight. Single colonies growing on the agar plates were selected and the elimination of the plasmid was examined by using both PCR and real-time quantitative PCR (qPCR) with primers targeting the IncFII(S) type replicons (F: 5'-CTGTCGTAAGCTGATGGC-3', R: 5'- CTCTGCCACAAACTTCAGC-3'). If the PCR result is still positive for the IncFII(S) plasmid replicons, the above-mentioned bacterial subculture in the EB-containing medium should be performed until the plasmid was eliminated successfully.

In plasmid transformation assays, a fragment in a non-coding area of plasmid pSH17G0407 was replaced using a kanamycin resistance gene from the plasmid pKD4 (GenBank accession no. AY048743) to generate a plasmid with kanamycin tag (pSH17G0407^kan). Thereafter, pSH17G0407^kan was transformed back into the plasmid-curing strains (SH17407M, SH17406M) to generate the complementary bacteria (SH17407R, SH17406R) through electroporation (200 OHMs, 25 uF, 2 kV). The stability of pSH17G0407^kan in complementary bacteria was assessed through bacterial passaging assays performed in LB broth containing 50 $\mu$g/mL kanamycin.

**Cell invasion assay, mouse challenge assays, and ethic statement.** To facilitate the analyses of invasion assays, Caco-2 cells were cultured in Dulbecco's modified eagle medium ([DMEM], Thermo Fisher) supplemented with 10% (vol/vol) heat- inactivated fetal bovine serum ([FBS], Gibco). Cells were

seeded into 12-well plates ($10^6$ cells per well) and cultured overnight. For bacterial preparation, overnight culture of plasmid elimination strains and their wild-type strains, as well as the complementary bacteria, were transformed into fresh LB broth at 1: 100 (vol/vol), and incubated at 37°C to $OD_{600}$ = 1.0. After centrifugation at 4°C, 6000 rpm for 5 min, bacterial pellets were harvested and were washed three times with phosphate-buffered saline (PBS), followed by resuspension in DMEM. Each well of the cells were inoculated with either the plasmid elimination strains, the wild-type strains, or the complementary bacteria or *S. Typhimurium* virulent ST19 strain SL1344 (58) at a multiplicity of infection (MOI) value of 100. After incubation at 37°C for 2 h, the cells were washed three times using PBS to remove the dissociative bacteria. Gentamicin (100 mg/mL) was given and the cells were incubated at 37°C for 1 h to kill bacteria adhered on the cell surface. Thereafter, cells were lysed using Triton X-100 buffer. A series of 10-fold dilutions were performed to the lysed cells using PBS, and appropriate dilutions were plated on LB agars. The agar plates were cultured at 37°C overnight for bacterial count.

Mouse tests were performed at the Laboratory Animal Center, Guangdong Academy of Agricultural Sciences (Guangzhou, China), and were approved by the Institutional Ethics Committee (IEC). Experimental animals were housed in the same environment and carried out under the guidelines established by the China Regulations for the Administration of Affairs Concerning Experimental Animals (1988) and Regulations for the Administration of Affairs Concerning Experimental Animals in Guangdong province (2010). Animal experiments were conducted following the protocols described previously (58). Briefly, SPF mice (C57BL/6, 15 to 17 g, purchased from the Guangdong Medical Laboratory Animal Center, Guangzhou, China) were divided into 7 groups (G1 to G7), and each group contained 5 mice. Prior to the challenge, mice received no water and food for 4 h. After that, the mice in G1, G2, and G3 were inoculated with S. Weltevreden strains SH170407, SH170406, and LS-2897 at a dose of $4 \times 10^6$ CFU (10 $\mu$L) by gavage, while each of the mice in G4, G5, and G6 were inoculated with the plasmid-curing mutants SH170407M, SH170406M, and LS-2897M at a dose of $4 \times 10^6$ CFU (10 $\mu$L) through the same routine, respectively. As a control, each of the mice in G7 received an administration of PBS (10 $\mu$L) by gavage. At 2 h postchallenge, water and food were given to the experimental mice. Body weights of the mice were recorded once every day for 7 days postchallenge. At 7 days postchallenge, mice were euthanized, and murine livers and spleens were collected for bacterial isolation and identification. Bacterial load was calculated as bacterial CFU recovered in one-gram (CFU/g) tissues.

**Statistical analysis.** Statistical analysis was performed using the multiple *t* tests strategy in GraphPad Prism8.0. Data represents mean $\pm$ standard deviation (SD). The significance level was set at $P < 0.05$ (*) or $P < 0.01$ (**).

**Data availability.** Whole-genome sequences of the 158 *S. enterica* serovar Weltevreden strains obtained in this study were deposited in GenBank with a BioProject ID PRJNA673740. Accession numbers for each of the genome sequences deposited are listed in Data sets S2 and 5. The complete genome sequence of the plasmid harbored in strain SH17G0407 was also deposited in GenBank under accession no. MW405382.

## SUPPLEMENTAL MATERIAL

Supplemental material is available online only.

**SUPPLEMENTAL FILE 1**, XLSX file, 0.01 MB.
**SUPPLEMENTAL FILE 2**, XLSX file, 0.01 MB.
**SUPPLEMENTAL FILE 3**, XLSX file, 0.02 MB.
**SUPPLEMENTAL FILE 4**, XLSX file, 0.01 MB.
**SUPPLEMENTAL FILE 5**, XLSX file, 0.01 MB.
**SUPPLEMENTAL FILE 6**, XLSX file, 0.05 MB.
**SUPPLEMENTAL FILE 7**, XLSX file, 0.4 MB.
**SUPPLEMENTAL FILE 8**, PDF file, 1.2 MB.

## ACKNOWLEDGMENTS

This work was supported by the National Key R&D Program of China (2022YFD1800400, 2017YFC1600101); National Natural Science Foundation of China (31972762); Guangdong Province Universities and Colleges Pearl River Scholar Funded Scheme (2018); Pearl River S&T Nova Program of Guangzhou (201806010183); Province Science and Technology of Guangdong Research Project (2017A020208055); Guangdong Key S&T Program (Grant no. 2019B020217002) from Department of Science and Technology of Guangdong Province; Walmart Foundation (Project # 61626817 & SA1703162) and supported by Walmart Food Safety Collaboration Center; National Broiler Industry Technology System Project (cARS-41-G16). The funders have no role in the study design, data collection and interpretation, or the decision to submit the work for publication.

We declare that there are no conflicts of interest or financial conflicts to disclose.

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
