## [Reviewer comments · Microbiology Spectrum]

Microbiology Spectrum

Genomic characterization of *Salmonella enterica* serovar Weltevreden associated with human diarrhea

Jianmin Zhang, Zhong Peng, Kaifeng Chen, Zeqiang Zhan, Haiyan Shen, Saixiang Feng, Hongchao Gou, Xiaoyun Qu, Mark Ziemann, Daniel Layton, Xiangru Wang, Huanchun Chen, Bin Wu, Xuebin Xu, and Ming Liao

Corresponding Author(s): Ming Liao, South China Agricultural University College of Veterinary Medicine

Review Timeline:

Submission Date:	September 2, 2022
Editorial Decision:	October 21, 2022
Revision Received:	December 3, 2022
Editorial Decision:	December 19, 2022
Revision Received:	December 20, 2022
Editorial Decision:	December 23, 2022
Revision Received:	December 28, 2022
Accepted:	January 4, 2023

Editor: Matthew Anderson

Reviewer(s): Disclosure of reviewer identity is with reference to reviewer comments included in decision letter(s). The following individuals involved in review of your submission have agreed to reveal their identity: Min Yue (Reviewer #2)

Transaction Report:

DOI: <https://doi.org/10.1128/spectrum.03542-22>

October 21, 2022

Prof. Ming Liao
South China Agricultural University
College of Veterinary Medicine
483 Wushan Road
Guangzhou, Guangdong
China

Re: Spectrum03542-22 (Genomic characterization of *Salmonella enterica* serovar Weltevreden associated with human diarrhea)

Dear Prof. Ming Liao:

Please see the reviews provided by two experts in the field. Please pay special attention to concerns about only some genomes being available for others to access, questions around experiments provided by Reviewer #2, and the transformation experiment proposed by Reviewer #1.

Link Not Available

Sincerely,

Matthew Anderson

Journals Department
Reviewer comments:

Reviewer #1 (Comments for the Author):

the authors have addressed most of the concerns by the reviewers. However, some questions remain. A specific one is covered in lines 260-262, the authors only tried to conjugate the plasmids back into the parental strains, because that failed, they could try to simply transform the plasmids into the parental strains. The authors could consider doing this additional experiment.

Reviewer #2 (Comments for the Author):

Zhang and colleagues (Spectrum03542-22) presented an investigation on an emerging Salmonella serovar, previous epidemiology investigations suggest a possible water-related source, however, very few studies were reported in China, and the newly provided genomic data in this study would be a value add to the global dataset. However, there are some concerns that should be improved in the revised version.

Line 29. how do you define "a comprehensive analysis", did you use all the data available in the global picture? or you did the genomic analysis other than AMR and VFs. There are many genetic factors or virulence factors that can be analyzed while it is missing in this study. I would be very careful by using this word.

Abstract, The only output I get is the ST365, what are the exact findings in your study, you should provide interesting or novel points here. And what are the implications of your investigation, how to improve the control of the serovar and ST...

Line 50, did you study the biogeography of this serovar of the world, I did not see any findings across China or between countries.

Line 57, what did you mean by "positive"

Line 66, 70, 75, and other, "Salmonella bacteria"?

Lines 71-73, should be rewritten.

Results. the sub-title should be illustrated as the results in brief, not what you did.
Please rewrite all the subtitles.

Line 103, did you consider "and the residual food samples" as human samples, I think they are food samples, but related to the patients.

Line 107, you select 96 for WGS, but why and how this selection is needed, there should be a reason for that should be illustrated.

Line 109, 141 genomes are obviously selected from the database, but how to select these genomes should be reasonably documented. I believe most of this data is not published, it is deposited there.

Line 116, there is also a reason that should be stated here.

Line 163, is Caco-2 a large intestine cell? Did you use the right control strain for this invasion study, which strain are you comparing with?

Line 170, there are a variety of plasmids detected here, how the logical rational link to one plasmid type, I would conduct a pan-genome analysis here to further understand the genetic diversity of this serovar.

Line 344, Only 58 genomes are available, in BioProject ID PRJNA673740.

Line 368, what are the detection limits for your PCR method? and what is the copy number for this IncFII(S) type plasmid?

Line 418, did you use only one type of statistic analysis, I think several different experiments were used, and the mice infection experiment should be different.

References, in general, are very old and should be updated with relevant and recent (within 5 years) work in the field.

Finally, the languages should be checked and improved by English-speaking experts, for readability and logical flow of the study.

Staff Comments:

Preparing Revision Guidelines

To submit your modified manuscript, log onto the eJP submission site at <https://spectrum.msubmit.net/cgi-bin/main.plex>. Go to Author Tasks and click the appropriate manuscript title to begin the revision process. The information that you entered when you first submitted the paper will be displayed. Please update the information as necessary. Here are a few examples of required

updates that authors must address:

Please return the manuscript within 60 days; if you cannot complete the modification within this time period, please contact me. If you do not wish to modify the manuscript and prefer to submit it to another journal, please notify me of your decision immediately so that the manuscript may be formally withdrawn from consideration by Microbiology Spectrum.

Reviewer comments:

Reviewer #1 (Comments for the Author):

The authors have addressed most of the concerns by the reviewers. However, some questions remain. A specific one is covered in lines 260-262, the authors only tried to conjugate the plasmids back into the parental strains, because that failed, they could try to simply transform the plasmids into the parental strains. The authors could consider doing this additional experiment.

We are grateful to your good comments and thank you for the useful suggestions. We have transformed the plasmid back in to the plasmid-curing strains through electroporation (lines 387-394), and have compared the invasion of plasmid-curing strains and plasmid-complementary strains to Caco-2 cells (lines 401, 405). The results demonstrated that plasmid-complementary strains displayed a stronger capacity of invasion than the plasmid-curing strains (lines 191-194, figure 4F).

This is a really wonderful suggestion, and thank you very much for that.

Reviewer #2 (Comments for the Author):

Zhang and colleagues (Spectrum03542-22) presented an investigation on an emerging Salmonella serovar, previous epidemiology investigations suggest a possible water-related source, however, very few studies were reported in China, and the newly provided genomic data in this study would be a value add to the global dataset. However, there are some concerns that should be improved in the revised version.

Thank you very much for your comments and suggestions. Below please find the places for revision in the revised manuscript responded by us point-by-point:

Line 29. how do you define "a comprehensive analysis", did you use all the data available in the global picture? or you did the genomic analysis other than AMR and VFs. There are many genetic factors or virulence factors that can be analyzed while it is missing in this study. I would be very careful by using this word.

Thank you for pointing this out for us. We have re-written this sentence. Please check lines 29 and 30 in the revised manuscript.

Abstract, The only output I get is the ST365, what are the exact findings in your study, you should provide interesting or novel points here. And what are the implications of your investigation, how to improve the control of the serovar and ST...

Good suggestions. We have rewritten the abstract part. Please check lines 29-46. Thank you.

Line 50, did you study the biogeography of this serovar of the world, I did not see any findings

across China or between countries.

We have reorganized this sentence and have removed the word “biogeography”. Please check line 50. As we stated in the original manuscript, we studied the sequence types, phylogenetic relationships, virulence factor genes, antimicrobial resistance genes, and plasmid replicons of this serovar from China and the other parts of the world based on the whole genome sequences. The findings are described in the whole result section (lines 97-197).

Line 57, what did you mean by "positive"

The results of cell invasion assays and mice challenging assays suggest that the plasmid might contribute to the virulence of the serovar. Thank you for your comments, and we have changed our statement here (lines 41-45).

Line 66, 70, 75, and other, "Salmonella bacteria"?

Thank you for pointing this out for us. We have revised this word in the whole manuscript. Please check lines 65, 69, 72, and the other places.

Lines 71-73, should be rewritten.

This sentence has been deleted from here as its deletion does not influence on our main findings and perspectives. Please check line 70. Thank you.

Results. the sub-title should be illustrated as the results in brief, not what you did.

Please rewrite all the subtitles.

Good suggestions. We have rewritten all subtitles in the result section following your suggestion. Please check them. Thank you.

Line 103, did you consider "and the residual food samples" as human samples, I think they are food samples, but related to the patients.

Thank you for pointing this out for us. We have revised our statement here. Please check line 101.

Line 107, you select 96 for WGS, but why and how this selection is needed, there should be a reason for that should be illustrated.

We isolated and stored 118 strains but only 96 ones were recovered from the stocks. This is why we only sequenced 96 strains. We have illustrated this reason here. Please check lines 103 and 104. Thank you.

Line 109, 141 genomes are obviously selected from the database, but how to select these genomes

should be reasonably documented. I believe most of this data is not published, it is deposited there.

Line 116, there is also a reason that should be stated here.

For these two concerns, the reason is there are only 199 assembled genome sequences, including 141 ones from humans and 58 ones from non-human species, in GenBank at the time of this paper writing (assessed as Nov 8, 2021; reference 20). We have added this reason in the manuscript as suggested. Please check lines 108-111. Thank you.

Line 163, is Caco-2 a large intestine cell? Did you use the right control strain for this invasion study, which strain are you comparing with?

Caco-2 cells (ATCC HTB-37) are epithelial cells isolated from colon tissue derived from a 72-year-old (<https://www.atcc.org/products/htb-37>). Thank you for pointing this out for us. We have corrected our statement here (line 164).

This control strain we used at here is the virulent *Salmonella* Typhimurium ST19 strain SL1344 (PMID: 12704158). This strain could lead to serious intestinal infections in both humans and animals. It exhibits particularly high levels of SPI1 expression, and it is one of the comparator strains used in previously published *in vitro* and *in vivo* studies (PMID: 26933058). Also, we have added the comparator strain at this place. Please check line 166.

Thank you very much.

Line 170, there are a variety of plasmids detected here, how the logical rational link to one plasmid type, I would conduct a pan-genome analysis here to further understand the genetic diversity of this serovar.

We focus on this plasmid because it is distributed in 95.04% (268/282) of the plasmid-carrying *S. Weltevreden* strains analyzed in this study (line 173-174). It is known that plasmids in members of *Enterobacteriaceae* play important roles in bacterial antimicrobial resistance and virulence because many antimicrobial resistance genes and virulence factor genes are disseminated by plasmids (e.g., PMID: 33298160; PMID: 33172969). We therefore focus on the analysis of this plasmid. Thank you.

Line 344, Only 58 genomes are available, in BioProject ID PRJNA673740.

Thank you for your comments. We have sent emails to NCBI for requesting the release of the remaining genome sequences.

Line 368, what are the detection limits for your PCR method? and what is the copy number for this IncFII(S) type plasmid?

We verified the elimination of the plasmid using two PCR methods. Using the ordinary PCR

method there is no band for the target gene and using the qPCR, there is no Ct value for the detection. However, we think the detection results did not affect the results because after the plasmid curing from the bacteria, their capacity of invading Caco-cells and mice decreased; however, after the plasmid being transferred back into the plasmid-curing strains, an increased capacity of invading to Caco-2 cells was observed compared to the plasmid-curing bacteria. Please see the results in lines 189-197, and figure 4. Thank you very much.

Line 418, did you use only one type of statistical analysis, I think several different experiments were used, and the mice infection experiment should be different.

Only “Multiple *t* tests” was used to compared data from different groups. After read your comments we have tried different methods on GraphPad Prism 8.0 and the results are similar. By searching published articles, we found statistical analysis based on “Multiple *t* tests” is acceptable (e.g., PMID: 36434990; PMID: 34686117; PMID: 34210738; PMID: 34408875; etc.). Thank you.

References, in general, are very old and should be updated with relevant and recent (within 5 years) work in the field.

Thank you for pointing this out for us. As we stated in the introduction section, *Salmonella enterica* serovar Weltevreden is a recently emerged pathogen. Therefore, there are not so many published articles. We have searched the key word “*Salmonella enterica* serovar Weltevreden” in PubMed and only 77 publications are obtained as of Nov 29, 2022, and only 23 of them are published within the recent five years. Notably, not all these 23 articles are related to the topic mentioned in our manuscript. On the contrary, many articles associated with the topics are published five years ago. This is why many citations are not very recent. Thank you.

Finally, the languages should be checked and improved by English-speaking experts, for readability and logical flow of the study.

Thank you for the suggestion. We have revised the manuscript carefully. Actually, before submission this manuscript has been edited by our two authors from Australia carefully, both of them are native speak with relevant academic background.

December 19, 2022

Prof. Ming Liao
South China Agricultural University College of Veterinary Medicine
College of Veterinary Medicine
483 Wushan Road
Guangzhou, Guangdong
China

Re: Spectrum03542-22R1 (Genomic characterization of *Salmonella enterica* serovar Weltevreden associated with human diarrhea)

Dear Prof. Ming Liao:

Thank you for submitting your manuscript to Microbiology Spectrum. As you will see your paper is very close to acceptance. Please modify the manuscript along the lines I have recommended:

- Please pay special attention to the points of readability of the manuscript in its current form. I might suggest a colleague who is especially strong in English grammar or a service for grammatical context.
- Please also include the context for *Salmonella* in China as noted by Reviewer #2.

As these revisions are quite minor, I expect that you should be able to turn in the revised paper in less than 30 days, if not sooner. If your manuscript was reviewed, you will find the reviewers' comments below.

When submitting the revised version of your paper, please provide (1) point-by-point responses to the issues raised by the reviewers as file type "Response to Reviewers," not in your cover letter, and (2) a PDF file that indicates the changes from the original submission (by highlighting or underlining the changes) as file type "Marked Up Manuscript - For Review Only". Please use this link to submit your revised manuscript. Detailed instructions on submitting your revised paper are below.

Link Not Available

Sincerely,

Matthew Anderson

Reviewer comments:

Reviewer #1 (Comments for the Author):

The authors have addressed my previous concerns about the manuscript.

Reviewer #2 (Comments for the Author):

There are a few text and clarifications issues

Abstract

"Continuous studying the genomics and epidemiology of the emerged pathogen *Salmonella* Weltevreden is necessary." Why is it necessary, there should be a context of question(s).

"However, we find *S. Weltevreden* strains carried a mass of virulence factor genes, and a 100.03-kb IncFII(S) type plasmid was widely distributed in *S. Weltevreden* strains." It is not However, it should be "Interestingly".

Some new references regarding Salmonella and Salmonellosis should be added. A big picture regarding the prevalence and disease burden in China should be the ideal contextual information for this study, even though this server is rarely studied in China. In other words, a more recent and comprehensive literature summary is needed in the introduction and discussion.

Preparing Revision Guidelines

Please return the manuscript within 60 days; if you cannot complete the modification within this time period, please contact me. If you do not wish to modify the manuscript and prefer to submit it to another journal, please notify me of your decision immediately so that the manuscript may be formally withdrawn from consideration by Microbiology Spectrum.

Reviewer comments:

Reviewer #1 (Comments for the Author):

The authors have addressed my previous concerns about the manuscript.

Thank you very much for your nice comments and suggestions, which would of course improve our manuscript remarkably.

Merry Christmas and happy the coming New Year.

Reviewer #2 (Comments for the Author):

There are a few text and clarifications issues

We do really appreciate your work and comments on our work and manuscript. Blew please find the places for revision in the revised manuscript responded by us point-by-point:

Abstract

"Continuous studying the genomics and epidemiology of the emerged pathogen Salmonella Weltevreden is necessary." Why is it necessary, there should be a context of question(s).

We have rephrased this sentence as “Salmonella Weltevreden is an emerging pathogen associated with human diarrhea and knowledge of the genomics and epidemiology of this serovar is still limited”. Please see lines 29-30.

"However, we find S. Weltevreden strains carried a mass of virulence factor genes, and a 100.03-kb IncFII(S) type plasmid was widely distributed in S. Weltevreden strains." It is not However, it should be "Interestingly".

Thank you for pointing this out and we have replaced the word “However” with “Interestingly”. Thank you. See line 42.

Some new references regarding Salmonella and Salmonellosis should be added. A big picture regarding the prevalence and disease burden in China should be the ideal contextual information for this study, even though this server is rarely studied in China. In other words, a more recent and comprehensive literature summary is needed in the introduction and discussion.

This is really a good suggestion. We have included more recent articles (most of them published after 2018) reflecting the detection and prevalence of Salmonella in China. Please check lines 68-72, and 78-87.

The whole manuscript has been re-edited by our two-coauthors from Australia (Dr Ziemanne and Dr Layton) carefully. All of them are native speaker with professional

experience in Microbiology and Infectious Diseases. Those revised places are also marked in GREEN in the revised manuscript.

Thank you very much again.

Merry Christmas and happy the coming New Year.

December 22, 2022

Prof. Ming Liao
South China Agricultural University College of Veterinary Medicine
College of Veterinary Medicine
483 Wushan Road
Guangzhou, Guangdong
China

Re: Spectrum03542-22R2 (Genomic characterization of *Salmonella enterica* serovar Weltevreden associated with human diarrhea)

Dear Prof. Ming Liao:

Thank you for submitting your manuscript to Microbiology Spectrum. As you will see your paper is very close to acceptance. There has been a large improvement in the readability of the manuscript and the inclusion of the reviewer recommended material has accomplished exactly what was hoped for. However, there are still many grammatical mistakes throughout the text, more than I feel comfortable accepting. Please resend the manuscript to your collaborators if needed as they missed many errors I found that include the following:

Line 111-112: rephrase as "...from diarrheal stool and blood..."

Line 119: "determined as" should be "determined to be"

The sentence on lines 158-163 needs to be reworked.

Line 167 and 168: "resistant" should be "resistance"

Line 172: "and these VFGs were"

Line 176 "of" should be "for"

Line 178 "where as" should be "whereas"

Line 205 "to" should be "of"

Line 206: "of" should be "with"

Line 207: "more" should be "greater"

Please modify the manuscript along the lines I have recommended. As these revisions are quite minor, I expect that you should be able to turn in the revised paper in less than 30 days, if not sooner. If your manuscript was reviewed, you will find the reviewers' comments below.

When submitting the revised version of your paper, please provide (1) point-by-point responses to the issues raised by the reviewers as file type "Response to Reviewers," not in your cover letter, and (2) a PDF file that indicates the changes from the original submission (by highlighting or underlining the changes) as file type "Marked Up Manuscript - For Review Only". Please use this link to submit your revised manuscript. Detailed instructions on submitting your revised paper are below.

Link Not Available

Sincerely,

Matthew Anderson

Reviewer comments:

Preparing Revision Guidelines

Please return the manuscript within 60 days; if you cannot complete the modification within this time period, please contact me. If you do not wish to modify the manuscript and prefer to submit it to another journal, please notify me of your decision immediately so that the manuscript may be formally withdrawn from consideration by Microbiology Spectrum.

Comments:

Thank you for submitting your manuscript to Microbiology Spectrum. As you will see your paper is very close to acceptance. There has been a large improvement in the readability of the manuscript and the inclusion of the reviewer recommended material has accomplished exactly what was hoped for. However, there are still many grammatical mistakes throughout the text, more than I feel comfortable accepting. Please resend the manuscript to your collaborators if needed as they missed many errors I found that include the following:

The whole manuscript has been re-edited again by our two-coauthors from Australia (Dr Ziemanne and Dr Layton) carefully. All of them are native speaker with professional experience in Microbiology and Infectious Diseases. Those revised places are also marked in BLUE in the revised manuscript.

Line 111-112: rephrase as "...from diarrheal stool and blood..."

We have rephrased this place as suggested. Please see lines 111-112. Thank you.

Line 119: "determined as" should be "determined to be"

"...determined as" has been revised into "determined to be", see line 119.

The sentence on lines 158-163 needs to be reworked.

We have reworked this sentence as suggested. Please see lines 158-163. Thank you.

Line 167 and 168: "resistant" should be "resistance"

This place has been revised as suggested. See line 168. Thank you.

Line 172: "and these VFGs were"

We have revised this place as suggested. Please see line 173.

Line 176 "of" should be "for"

We have changed "of" to "for" as suggested. See line 177.

Line 178 "where as" should be "whereas"

Sorry for the typo. We have revised this place. See line 178.

Line 205 "to" should be "of"

We have revised this place as suggested. See line 205.

Line 206: "of" should be "with"

We have revised this place as suggested. See line 206.

Line 207: "more" should be "greater"

We have revised this place as suggested. See line 207.

Thank you very much again.

January 4, 2023

Prof. Ming Liao
South China Agricultural University College of Veterinary Medicine
College of Veterinary Medicine
483 Wushan Road
Guangzhou, Guangdong
China

Re: Spectrum03542-22R3 (Genomic characterization of *Salmonella enterica* serovar Weltevreden associated with human diarrhea)

Dear Prof. Ming Liao:

Your manuscript has been accepted, and I am forwarding it to the ASM Journals Department for publication. You will be notified when your proofs are ready to be viewed.

In the proofs, please address some small text edits:

Line 72: insert "and" after "2017"

Line 97: the "," before however should be a ";"

Line 257: insert "," before and after the phrase "including the ST365 clone"

Sincerely,

Matthew Anderson
Editor, Microbiology Spectrum
